# Aberrant Mitochondrial Morphology and Function in the BTBR Mouse Model of Autism Is Improved by Two Weeks of Ketogenic Diet

**DOI:** 10.3390/ijms21093266

**Published:** 2020-05-05

**Authors:** Younghee Ahn, Rasha Sabouny, Bianca R. Villa, Nellie C. Yee, Richelle Mychasiuk, Golam M. Uddin, Jong M. Rho, Timothy E. Shutt

**Affiliations:** 1Departments of Pediatrics, Clinical Neurosciences, Physiology & Pharmacology, Alberta Children’s Hospital Research Institute, Hotchkiss Brain Institute, Cumming School of Medicine, University of Calgary, Calgary, AB T2N 4N1, Canada; yahn@ucalgary.ca (Y.A.); bianca.villa1@ucalgary.ca (B.R.V.); ncyee@ucalgary.ca (N.C.Y.); jrho@health.ucsd.edu (J.M.R.); 2Departments of Medical Genetics and Biochemistry & Molecular Biology, Alberta Children’s Hospital Research Institute, Cumming School of Medicine, Hotchkiss Brain Institute, University of Calgary, Calgary, AB T2N 4N1, Canada; Rasha.Sabouny@ucalgary.ca (R.S.); golammezbah.uddin@ucalgary.ca (G.M.U.); 3Department of Psychology, Alberta Children’s Hospital Research Institute, Hotchkiss Brain Institute, University of Calgary, Calgary, AB T2N 4N1, Canada; richelle.mychasiuk@monash.edu; 4Department of Neuroscience, Monash University, Melbourne, VIC 3004, Australia

**Keywords:** autism spectrum disorder, mitochondria, mitochondrial dynamics, fission, fusion, ketogenic diet, BTBR mouse, mitochondrial function

## Abstract

Autism spectrum disorder (ASD) is a highly prevalent neurodevelopmental disorder that exhibits a common set of behavioral and cognitive impairments. Although the etiology of ASD remains unclear, mitochondrial dysfunction has recently emerged as a possible causative factor underlying ASD. The ketogenic diet (KD) is a high-fat, low-carbohydrate diet that augments mitochondrial function, and has been shown to reduce autistic behaviors in both humans and in rodent models of ASD. The aim of the current study was to examine mitochondrial bioenergetics in the BTBR mouse model of ASD and to determine whether the KD improves mitochondrial function. We also investigated changes in mitochondrial morphology, which can directly influence mitochondrial function. We found that BTBR mice had altered mitochondrial function and exhibited smaller more fragmented mitochondria compared to C57BL/6J controls, and that supplementation with the KD improved both mitochondrial function and morphology. We also identified activating phosphorylation of two fission proteins, pDRP1^S616^ and pMFF^S146^, in BTBR mice, consistent with the increased mitochondrial fragmentation that we observed. Intriguingly, we found that the KD decreased pDRP1^S616^ levels in BTBR mice, likely contributing to the restoration of mitochondrial morphology. Overall, these data suggest that impaired mitochondrial bioenergetics and mitochondrial fragmentation may contribute to the etiology of ASD and that these alterations can be reversed with KD treatment.

## 1. Introduction

Autism is a complex, heterogeneous neurodevelopmental disorder characterized by abnormal social interactions, communication deficits, and repetitive movements [1,2,3,4]. A recent study suggested a 2.5% prevalence of autism spectrum disorder (ASD) in the United States [5]. While the etiology of ASD remains unknown, it is likely that a combination of genetic, epigenetic, and environmental risk factors are involved [6]. Moreover, many children with ASD present with medical comorbidities, such as epilepsy [7], gastrointestinal (GI) abnormalities [8], and mitochondrial disease [2,9,10,11,12].

Since Lombard postulated that ASD may be caused by mitochondrial dysfunction in 1998 [13], there are growing lines of evidence from clinical, genetic and biochemical studies supporting a role for mitochondrial dysfunction in ASD [14,15,16]. Estimates suggest that 5–80% of ASD patients exhibit markers of mitochondrial dysfunction [16]. Similarly, variations in mitochondrial DNA sequence and copy number are also associated with ASD [17,18]. Finally, toxins linked to ASD also inhibit mitochondrial function [19,20].

There is an established association between mitochondrial function and mitochondrial shape, with dysfunctional mitochondria typically exhibiting a smaller more fragmented morphology [21]. Notably, changes in mitochondrial function can influence mitochondrial shape, while the reverse is also possible (i.e., changes in shape can affect mitochondrial function). Although decreased mitochondrial function and increased mitochondrial fragmentation have been reported in ASD patients [22], it is unclear what induces these alterations. Also at question is the relevance of these changes to ASD pathology. If mitochondrial dysfunction contributes to the development of ASD, then restoring function, either by targeting oxidative phosphorylation or mitochondrial morphology, could be beneficial and potentially may lead to better therapeutic approaches.

In the current study, we used BTBRT^+tf/j^ (BTBR) mice as a representative mouse model of ASD. The BTBR mouse exhibits a robust ASD phenotype, characterized by three core symptoms; impaired sociability, communication deficits, and repetitive/stereotypic behaviors [23]. Importantly, C57BL/6J mice are routinely used as the control strain for studies using BTBR mice [24,25]. Notably, a number of spontaneous mutations have occurred since the divergence of these strains, which could be relevant to mitochondrial function. For example, there is a deletion in the *ITPR3* gene encoding the type 3 inositol 1,4,5- triphosphate receptor in the BTBR strain [26]. Given that IP3 receptors can mediate interactions between mitochondria and the endoplasmic reticulum [27], this deletion could potentially have an effect on mitochondrial function in BTBR mice.

The ketogenic diet (KD) is a high-fat, low-carbohydrate and low-protein diet designed to switch the primary source of cellular energy from glucose to fatty acids [28] and is known as a remarkably effective non-pharmacological treatment for medically intractable epilepsy [29]. Importantly the KD has been linked to improved ASD behaviors [30,31]. Specifically, multiple case reports [32,33] and small scale studies [34,35,36,37], report benefits of the KD. In addition, preclinical studies have shown that the KD reduces ASD behaviors in multiple rodent models of ASD [38,39,40,41,42], including BTBR mice [43]. While the KD has clear effects on mitochondrial function [44] and has been shown to promote elongation of the mitochondrial network [45], the underlying mechanisms have not been determined. Here, we examined the effects of the KD on mitochondrial function and dynamics in BTBR mice and age-matched control mice, to investigate if and how the KD improves mitochondrial abnormalities in the BTBR model of ASD.

## 2. Results

As short term (2-3 weeks) administration of the KD improves ASD behaviors in BTBR mice [42,45], we decided to replicate previous KD treatments, but instead focus here on mitochondrial function.

### 2.1. Two Weeks of KD Significantly Reduced Mice Body Weight and Plasma Glucose Levels

In addition to the KD’s ability to switch the primary source of cellular energy [28], it has the potential to reduce body weight [46]. We found that compared to mice fed the SD, body weight for both control and BTBR mice was reduced (Figure 1B) after a week of KD supplementation, and that this weight loss persisted at two weeks (Figure 1B, right panel). Notably, there were no adverse health effects in these animals that might be associated with the weight loss. To confirm that the diet was indeed inducing ketosis in these mice, we measured the plasma ketone levels after 2-weeks of KD. We found that mice fed the KD had significantly increased ketone levels (Figure 1C). Additionally, we found a significant reduction of plasma glucose levels due to KD in both groups of mice (Figure 1D), a further indicator of a metabolic shift in the KD-fed animals. 

### 2.2. KD Increased Mitochondrial Metabolism and Decreased AMPK Activation in BTBR Mice

Given the established links between ASD and mitochondrial dysfunction, we initially evaluated mitochondrial function in the BTBR mouse brain (Figure 2). We tested mitochondrial respiration using mitochondria isolated from BTBR and control mouse brains at PD35. This approach allowed us to investigate mitochondrial function at an age when ASD behaviors manifest and enabled us to interrogate potential molecular mechanisms mediating the beneficial effects of the KD. The purity of mitochondrial fraction was confirmed by Western blotting (Appendix A). Consistent with previous studies in ASD patients [22], we found a decreased basal oxygen consumption rate (OCR) in mitochondria isolated from BTBR mice, compared to control mouse brain (Figure 2A). Mitochondria isolated from BTBR mice also exhibited lower relative ATP production, calculated from ATP-linked respiration (Figure 2B). Notably, the KD significantly increased basal OCR in control mice, and restored basal OCR in the BTBR mice compared to control mice on the SD (Figure 2A). In addition, relative ATP production also increased with the KD, although differences were not statistically different (Figure 2B).

AMP-activated protein kinase (AMPK) is an energy sensor that is activated when ATP levels are reduced and AMP levels increase. Given the trends observed for the relative ATP production observed in BTBR mice (Figure 2B), we decided to evaluate AMPK activity. We found increased phosphorylation of AMPK^T172^ in BTBR mice, consistent with AMPK activation [47]. Moreover, the KD significantly reduced phosphorylation of AMPK^T172^ (ratio) in BTBR mice, with a trend towards reduced AMPK phosphorylation in controls (Figure 2C). This result is consistent when pAMPK^T172^ expression was normalized to the actin loading control (Figure 2D, Appendix A). Thus, these results indicate that the KD restores the aberrant activation of AMPK in BTBR mice, possibly by increasing ATP production. Interestingly, when we analyzed the total AMPK expression, we found an elevated AMPK expression only in the control group due to KD (Figure 2D, Appendix A).

### 2.3. BTBR Have Increased Complex II Enzyme Activity and Protein Expression

The standard approach to measure OCR (Figure 2A,B), uses pyruvate and malate, which are substrates that feed into Complex I. To determine whether the decreased OCR was due to alterations in Complex I or other downstream complexes, we examined Complex II activity. To this end, the addition of a Complex I inhibitor (rotenone, 2µM), and a Complex II substrate (succinate, 5 mM), allowed us to bypass Complex I activity and measure flux through Complex II. In this modified assay for Complex II activity, isolated mitochondria from BTBR mice showed a significantly higher OCR than the mitochondrial from control mice, which was reversed by the KD (Figure 2E). These data suggest that there are compensatory increases in Complex II activity in response to decreased Complex I activity. 

To validate the increased Complex II activity observed via Seahorse, we measured Complex II enzyme activity using a different assay (Figure 2F) and also observed increased activity in BTBR mice, which was restored to control levels by the KD. Next, to investigate the mechanism responsible for changing Complex II activity, we checked the abundance of Complex II proteins. Curiously, while Complex II protein expression levels were increased in BTBR mice, consistent with the increased Complex II activity we observed, the KD did not affect the expression of Complex II proteins (Figure 2G). This finding implies that although there is an increase in the expression of Complex II proteins in the BTBR mice, the beneficial effects of the ketogenic diet are not due to changes in the expression of Complex II proteins. 

### 2.4. KD Rescues Mitochondrial Fragmentation Observed in Brains of BTBR Mice

As mitochondrial fragmentation, which is typically associated with mitochondrial dysfunction, has been reported in the brains of ASD patients [22], we wanted to examine mitochondrial shape in the brains of BTBR mice (Figure 3A and Appendix A). Thus, we measured mitochondrial size and numbers in the frontal cortex of control and BTBR mice using electron microscopy (Figure 3B). While no significant changes in the total number of mitochondria were observed, the mean length for mitochondrial size was significantly decreased in BTBR mice compared to control B6 mice. Importantly, in the BTBR mice, the KD elicited an increase in mitochondrial length compared to the SD (Figure 3C).

### 2.5. The KD Metabolite, β-hydroxybutyrate, Restores Mitochondrial Fragmentation Observed in Primary Neuronal Cells from BTBR Mice

In order to investigate how the KD might be affecting mitochondrial morphology in BTBR mice, we added β-hydroxybutyrate (βHB), a metabolite produced by the KD [48,49], to primary neuronal cortical cultures from control and BTBR mice harvested at P0 and cultured for five days. Using confocal microscopy, we examined the mitochondrial morphology of primary neuronal cultures using immunofluorescence with antibodies to the neuronal marker MAP2, and the mitochondrial outer membrane protein TOMM20. Mitochondrial morphology in neurites (dendrites) was classified into fragmented, intermediate and fused networks as described in Appendix A. As with our electron microscopy images, we observed more fragmented mitochondria in BTBR mice compared to control mice under standard growth conditions. Moreover, compared to the vehicle control group, βHB treatment elicited an increase in fused mitochondrial networks in primary neuronal cultures from both control and BTBR mice (Figure 4). These data indicate that βHB alone is sufficient to restore mitochondrial morphology in neuronal cultures from BTBR mice.

### 2.6. Changes in Mitochondrial Dynamics Proteins in BTBR Mouse Brains Fed the KD

Mitochondrial morphology is mediated by the balance between two opposing forces, fusion (elongation) and fission (fragmentation). Therefore, to investigate the molecular mechanisms mediating changes in mitochondrial morphology in the brain of the BTBR mice and in response to the KD, we examined the expression and phosphorylation status of several key protein regulators of mitochondrial fission and fusion (Figure 5 and Figure 6). DRP1 is a key regulator of mitochondrial fission that responds to numerous signaling pathways. Phosphorylation of DRP1 in particular provides an important means of regulating DRP1 activity [50,51], with DRP1^S616^ phosphorylation activating fission, and DRP1^S637^ phosphorylation inhibiting fission. Interestingly, while total DRP1 expression, and levels of the inhibitory pDRP1^S637^ modification were not significantly altered (Figure 5C), we observed a significant increase in the levels of the activating pDRP1^S616^ modification in BTBR mice (Figure 5A,B). Moreover, KD treatment reduced the relative levels of DRP1^S616^ phosphorylation in BTBR mice. These findings are consistent with the observed changes in mitochondrial morphology and suggest that some of the benefits associated with the KD are due to alterations in mitochondrial morphology via reduced phosphorylation of DRP1^S616^.

Given that many players are involved in the regulation of mitochondrial morphology, we also examined other fusion/fission related proteins (Figure 6). The mitochondrial fission factor (MFF), and mitochondrial dynamics protein of 51kDa (MID51) are two receptors that recruit DRP1 to sites of fission [52]. However, the relationship and redundancy of these two adaptors is not completely understood. Nonetheless, previous studies showed that AMPK-mediated phosphorylation of MFF^S155^ and ^S172^ increased the recruitment of DRP1 to mitochondria [53,54,55]. Given the increase in AMPK signaling (Figure 2C), we investigated the levels of pMFF ^S146^ (the murine equivalent to human pMFF ^S172^). Notably, pMFF^S146^ was significantly increased in BTBR mice compared to control mice (Figure 6A,B), consistent with increased fission and smaller mitochondria. However, pMFF^S146^ levels in BTBR mice were not reduced by the KD (Figure 6A,B), despite a decrease in AMPK activity (Figure 2C). Meanwhile, MID51 protein levels were decreased in BTBR mice, but remained unchanged by the KD (Figure 6B,D).

The dynamin-related GTPase optic atrophy 1 (OPA1) is essential for mitochondrial fusion and is regulated by post-translational processing. We measured total OPA1 protein, as well as the ratio of long to short OPA1 isoforms (Figure 6B,C). While OPA1 protein levels were not different between control and BTBR mice, the KD did show a trend toward increased total OPA1 protein in BTBR mice, consistent with increased fusion. No significant changes in the ratio of long and short forms of OPA1 were observed. These results demonstrate that mitochondrial fragmentation in BTBR mice may be regulated by the recruitment of DRP1 by MFF and MID51, while mitochondrial elongation by the KD in the BTBR mice may be due to decreased phosphorylation of DRP1^S616^, and possibly increased expression of OPA1.

## 3. Discussion

There is a strong correlation between mitochondrial dysfunction and ASD [11]. For example, mitochondrial disease, which is caused by severe mitochondrial dysfunction, is much more prevalent in ASD patients (5.0%, compared to ~0.2% in the general population). Moreover, mild mitochondrial dysfunction is the most common metabolic abnormality associated with ASD [11,56], and is found in up to 80% of ASD patients [57]. Together, these observations suggest that mitochondrial dysfunction is a key contributing factor to ASD. The KD, which has been linked to behavioral improvements in both ASD patients [30,31] and multiple rodent models of ASD [38,39,40,41,42,43], is also known to improve mitochondrial function [44,45,46,47,48,49,50,51,52,53,54,55,56,57,58], further strengthening the correlation between mitochondrial dysfunction and ASD. However, the mechanisms responsible for the actions of the KD remain poorly understood.

This study begins to elucidate a mechanistic and functional link between mitochondrial bioenergetics and mitochondrial dynamics in the brain of BTBR mice fed either SD or KD. Notably, we observed different bioenergetics profiles in isolated mitochondria from BTBR and control mouse brains (P35), (Figure 2; Figure 3). The decreased mitochondrial activity in BTBR mouse brains is similar to what has been reported in ASD patient-derived lymphoblastoid cells [59]. In addition, consistent with reports in ASD patients [22], we also observed a more fragmented mitochondrial morphology in both brains (P35), and primary neuronal cortical cultures (P0) (Figure 3 and Figure 4). 

In further examining the decreases in oxidative phosphorylation in BTBR mitochondria, we observed decreased activity when using substrates feeding Complex I. This is relevant, as Complex I deficiencies are reported as the most common electron transport chain (ETC) disorders in autism [12]. It is notable that the KD is effective as a treatment for mitochondrial disorders due to Complex I impairment [60]. We also observed an increase Complex II activity in neuronal mitochondria of BTBR mice (Appendix A), which likely represents a compensatory increase due to impaired Complex I function. 

The decreased oxidative phosphorylation activity in BTBR mice may be linked to decreased relative ATP production (Figure 2B). These findings are consistent with the observation that children with autism have low levels of plasma ATP [61]. In addition, ASD patient plasma has lower biotin, decreased plasma glutathione, and higher levels of oxidative stress markers, all consistent with mitochondrial dysfunction. With this in mind, it is notable that the KD has the ability to both increase ATP levels and reduce reactive oxygen species [62,63]. Here, we also found that the KD restored the lower mitochondrial respiration and ATP levels in BTBR mice (Figure 2). Together with previously reported behavioral improvements of BTBR mice on the KD [43], these new data are consistent with the notion that decreased mitochondrial function underlies at least a subclass of ASD.

AMPK is a signaling kinase that senses the energy status of the cell to help maintain cellular energy homeostasis. Low levels of cellular ATP activate AMPK signaling and initiate a cellular stress response. In our hands, AMPK was activated in BTBR mice, which have lower relative ATP production. We observed that treatment of BTBR mice with the KD reduced AMPK activation (Figure 2C,D). These data suggest that the KD reduces aberrant activation of AMPK signaling in BTBR mice by restoring mitochondrial function and increasing ATP production.

Mitochondria are dynamic organelles that can fuse and divide. Balancing these opposing processes contributes to maintenance of not only mitochondrial shape, but also of mitochondrial and cellular homeostasis [64]. In general, compared to larger fused mitochondria, smaller more fragmented mitochondria are associated with lower ATP production and increased reactive oxygen species. Consistent with the decreased oxidative phosphorylation and decreased relative ATP production in BTBR mice, we also observed smaller mitochondria, which return to normal size after treatment with the KD or βHB (Figure 3 and Figure 4).

To better understand the mechanisms leading to these alterations in mitochondrial morphology, we examined several mediators of mitochondrial fission and fusion (Figure 5 and Figure 6). DRP1, a key regulator of mitochondrial fission, is recruited to mitochondria by adaptor proteins such as MFF and MID51. In BTBR mice relative to controls, we observed an increase in the ratio of DRP1^S616^ phosphorylation, a post-translational modification that activates mitochondrial fission. We further noted an increase in total levels of MFF and phosphorylation in BTBR animals, alterations known to enhance DRP1 recruitment and increase fission [55]. Importantly, MFF is phosphorylated by AMPK under low ATP levels [55]. While the total levels of MID51 were decreased in BTBR mice, it is unclear exactly how MID51 mediates changes in mitochondrial morphology, especially in the context of the other DRP1 recruiting proteins [65]. Nevertheless, the observed decrease in MID51 in BTBR mice may represent a compensatory response to increases in fission mediated by DRP1 and MFF phosphorylation.

We also observed changes in the profiles of fusion and fission proteins in BTBR mice fed the KD, providing novel insights into how the KD affects mitochondrial function. First, while we observed a decrease in the level of DRP1^S616^ phosphorylation in BTBR mice following KD treatment, we did not see any changes in the expression of MID51 or in the phosphorylation status MFF, two proteins involved in DRP1 recruitment. These findings suggest that changes in DRP1^S616^ levels may be more important than its recruiting proteins. Surprisingly, the MFF phosphorylation levels did not change, despite a decrease in AMPK activation, suggesting that there may be other unrecognized pathways regulating MFF phosphorylation in addition to AMPK. 

Finally, we observed a trend towards increased expression of the mitochondrial fusion protein OPA1 in BTBR mice fed the KD, which may also contribute to the restoration of mitochondrial morphology in these mice. One possible explanation for the increased expression of OPA1 is through peroxisome proliferator-activated receptor-γ coactivator-1α (PCG-1α) expression, which controls genes involved in mitochondrial biogenesis and dynamics [66]. Importantly, PGC-1α is increased by the KD (also through exercise or caloric restriction) [45,64], and thus may be an important regulator mediating the benefits of the KD.

While we have seen that the KD improves mitochondrial morphology in BTBR mice, there are certainly other ways by which the KD could have beneficial effects on mitochondrial function. For example, the KD is known the reduce ROS by upregulating the NRF2 antioxidant pathway [67]. In addition, the KD could upregulate mitochondrial autophagy, which is an important mitochondrial quality control pathway [68].

Mitochondrial dysfunction is increasingly recognized to play a role in a growing number of neurological disorders, such as ASD, and thus approaches that target mitochondrial dysfunction are of particular interest. Such approaches include the administration of antioxidants [69] and activators of mitochondrial biogenesis [70], which have been proposed for ASD [71]. However, these approaches have not been widely validated in clinical populations. In this regard, the KD, which is relatively safe and easy to implement, has been recognized for over a century for its application in epilepsy, and has also proven to be beneficial to a growing list of neurological disorders [72].

## 4. Materials and Methods

### 4.1. Animals and Dietary Procedure

BTBR mice were obtained from the Jackson Laboratory (Bar Harbor, ME, USA) and a breeding colony has been maintained at the Health Sciences Animal Resource Center at University of Calgary. Detail animal protocol is presented in the schematic diagram (Figure 1A). The mice were born and reared in a quiet, temperature-controlled room and entrained to a 12-hour light-dark cycle. After weaning at postnatal day 21 (PD21), the mice were placed on either standard diet (SD) or ketogenic diet (KD; 6.3:1 ratio by weight of fat to carbohydrate plus protein; Bio-Serv F3666, Frenchtown, NJ, USA) for 10-14 days [73,74] based on our previous experiments [38,45]. F3666 is a ketogenic chow composed of lard, butter, corn oil, casein, cellulose, mineral mix, vitamin mix, and dextrose. This particular fomulation is rich in saturated fats [75]. Body weight was measured weekly the KD treatment (Figure 1). After dietary intervention, whole blood was analyzed for glucose and circulating ketone bodies (β-hydroxybutyrate) with Precision Xtra meters (Abbott Laboratories, Chicago, IL, USA) before animals were sacrificed. All experiments were performed around PD35. Only male mice were used. All procedures were carried out in accordance with the Canadian Animal Care Committee and approved by the University of Calgary Animal Care Committee under the protocol AC17-0217.

### 4.2. Mitochondria Isolation and Protein Assay

Total mitochondria were isolated using differential centrifugation, nitrogen disruption, and a Ficoll gradient as previously reported [76,77,78]. Mice were euthanized using isoflurane and rapidly decapitated in accordance with approved IACUC protocols. Following decapitation, the brain was removed and placed briefly in a beaker of ice-cold isolation buffer (215 mM mannitol, 75 mM sucrose, 0.1% BSA, 1 mM EGTA, and 20 mM HEPES at pH 7.2). Neo-cortical brain tissues were homogenized and then centrifuged at 1300× *g* for 3 min at 4 °C. The supernatant was collected in a fresh tube and centrifuged at 13,000× *g* for 10 min at 4 °C. The pellet was resuspended in 600 μL isolation buffer and placed in a nitrogen bomb at 1200 psi for 10 min at 4°C. The pressure in the nitrogen bomb was released, and then the sample was placed as the top layer on a Ficoll separation column, which consisted of a 10% Ficoll layer and a 7.5% Ficoll layer. The Ficoll column with sample was centrifuged at 32,000 rpm for 30 min at 4 °C. Following the Ficoll purification, the mitochondrial pellet was resuspended in isolation buffer without EGTA and centrifuged at 10,000× *g* for 10 min at 4 °C to remove residual Ficoll from the purified mitochondrial sample. The final mitochondrial pellet was resuspended in an isolation buffer without EGTA to yield a final concentration of approximately 10 mg/mL and was stored immediately on ice. Protein concentration for a sample was determined using the BCA protein assay and measuring absorbance at 562 nm with a Synergy HT Plate Reader (BioTek, Winooski, VT, USA). The purity of the mitochondrial fraction was corroborated by Western blot with antibodies directed against calcium/calmodulin-dependent protein kinase I (CamKI, Abcam, 1:600), histone H3 (Abcam, 1:10,000), VDAC (Abcam, 1:5000), and cytochrome C (Santa Cruz, 1:300) (Appendix A).

### 4.3. Mitochondrial Bioenergetics Assay Using Seahorse XFe24

A Seahorse Bioscience XFe24 extracellular flux analyzer was used to measure mitochondrial function in cultured cells or intact isolated mitochondria. The day before an experiment, 1 mL of XF calibrant solution (Agilent, Santa Clara, CA, USA) was added to each well of a 24-well dual-analyte sensor cartridge (Agilent). The cartridge was placed back on the 24-well calibration plate and put overnight in a 37°C incubator without CO2. On the day of the experiment, injection ports on the sensor cartridge were loaded with specific mitochondrial substrates or inhibitors at 10× concentrations. Once loaded, the sensor cartridge was placed into the Seahorse XFe24 flux analyzer for automated calibration. The isolated mitochondria were diluted to the concentration required for plating (2.5 μg/50 μL) in respiration buffer (125 mM KCl, 2 mM MgCl2, 2.5 mM KH2PO4, 20 mM HEPES, and 0.1% bovine serum albumin (BSA) at pH 7.2) during calibration. Next, 50 μL of mitochondrial suspension was delivered to each well (except for background correction wells; A1, B3, C4, and D6) while the plate was on ice. The plate was then transferred to a centrifuge equipped with a swing bucket microplate adaptor, and was spun at 2000 rpm for 20 min at 4°C. After centrifugation, 450 μL of prewarmed respiration buffer at 37 °C was gently added to each well for a final volume of 500 μL per well at the beginning to the experiment. Plates were then immediately placed into the calibrated analyzer for mitochondrial bioenergetics analysis. ADP/pyruvate/malate (4/5/2.5 mM), oligomycin (2 μg/mL), FCCP (4 μM), and antimycin A (2 μM) were injected sequentially through ports A–D, respectively. 

### 4.4. Assay for Mitochondrial Electron Transport Chain (ETC) Complex II Enzyme Activity

ETC complex II enzyme activity was analyzed using an Abcam assay kit following the manufacturer’s instructions (Abcam, Cambridge, United Kingdom; ab109908). The isolated mitochondria from the neo-cortical brain were used for complex II enzyme activity (100 μg proteins). The final colorless reaction product for complex II enzyme activity was monitored by measuring the decreases in absorbance at 600nm by the Synergy HT microplate reader (BioTek, Winooski, VT, USA). To select the optimal amount for the activity assay, a dose-effect curve was plotted for the assay.

### 4.5. Primary Cortical Mixed Cell Culture

Primary cultures of cortical neuron and glial cells were prepared from whole cortices of PD0 BTBR or C57 mice. Cortices were dissected and dissociated with papain for 30 min, and then triturated with decreasing pore size pipettes. Cells were plated in growth media (33 mM glucose, 2 mM glutamine, 10 mM HEPES, 1 mM sodium pyruvate, 1X B27, 50 units/mL penicillin-streptomycin, and 4% fetal bovine serum (FBS)) on poly-D-lysine and laminin coated 12 mm coverslips and allowed to grow for 4 days at 37 °C and 5% CO_2_. At day 4, culture media was supplemented with the indicated concentrations of β-hydroxybutyrate (Sigma, St. Louis, MO, USA) or vehicle control (H2O). After 24 h, culture media was aspirated, and the cells were gently washed with prewarmed 1X PBS and fixed in prewarmed 4% paraformaldehyde for 15 min at 37°C. The fixative was subsequently aspirated, coverslips were washed 3 times with 1X PBS, quenched in 50 mM ammonium chloride for 15 min at room temperature and washed with 1X PBS.

### 4.6. Immunofluorescence Staining

Cells were permeabilized with 0.2% TritonX100 in PBS for 15 min at room temperature, washed with 1X PBS, and blocked in 10% FBS in PBS for 30 min. The following primary antibodies were used at 1:1000, diluted in 5% FBS in PBS, and cells were incubated at 37 °C for 1 h: TOMM20 (Santa Cruz, Dallas, TX, USA) and MAP2 (Signa-Aldrich, St Louis, MO, USA). The appropriate secondary Alexa Fluor antibodies were diluted in 5% FBS in PBS at 1:1000, and cells were incubated in the dark at room temperature for 1 h. Stained cells were then washed in 1X PBS and mounted using Dako mounting media (Agilent; S3023).

### 4.7. Microscopy and Mitochondrial Morphology Analysis

An Olympus spinning disc confocal system (Olympus SD OSR, Olympus Corporation, Tokyo, Japan) (UAPON 100XOTIRF/1.49 oil objective) operated by Metamorph software was used to image fixed neuronal cultures. Mitochondrial network morphology in primary cortical cultures was assessed in neurites (dendrites) of MAP2-positive neurons in a blinded fashion and assigned to 3 categories of mitochondrial morphology (fragmented, intermediate, and fused). At least 50 cells were counted per condition, and the analyses were performed on 3 independent replicates for each treatment condition.

### 4.8. Mitochondrial Number and Size Determination Using Transmission Electron Microscopy (TEM)

Mitochondrial size and number were assessed from micrographs for three tissue slices obtained from each of the three C57 and three BTBR mice, for a total of 9 micrographs per group. The length of all complete mitochondria on individual TEM micrographs was estimated by standard cross-sectional interpolation [79].

### 4.9. Western Blots

Depending on the target, 10–20-μg total protein was loaded onto Mini-PROTEAN^®^ Precast Gels (Bio-Rad, Hercules, CA, USA) and transferred to polyvinylidene fluoride (PVDF) membranes (Bio-Rad). Membranes were then blocked with 5% nonfat milk in Tris-buffered saline solution containing 0.1% Tween 20 (TBS-T) for 1 h. After blocking, the blots were incubated with primary antibody overnight at 4 °C. After washing with TBS-T, the blots were incubated with the secondary antibody for 1 h at room temperature. After washing with TBS-T, blots were visualized using an enhanced chemiluminescence (ECL) detection system. Bands were detected and quantified using a ChemiDOC MP gel imaging system (Bio-Rad). Antibodies against the Complex 1–5 cocktail (Abcam; 1:5000); DRP1 (BD Biosciences, San Jose, CA, USA; 1:2000); pDRP1 at S616 (Cell Signaling, Danvers, MA, USA; 1:1000); AMPK (Cell Signaling; 1:500); pAMPK at T172 (Cell Signaling; 1:500); MFF (ProteinTech, Rosemont, IL, USA; 1:5000); pMFF at S146 (Cell Signaling; 1:1000); OPA1 (BD Biosciences; 1:5000); and MID51 (ProteinTech; 1:1000) were used. VDAC (Abcam; 1:5000) or actin (Cell Signaling; 1:10,000) was used as a loading control for all experiments. The expression levels of proteins were normalized to VDAC for mitochondria or to actin for tissue lysate, and the ratio of the phosphorylated and total forms was computed using the normalized relative expression levels.

### 4.10. Statistics

The statistical analysis of the data was performed using GraphPad Prism 5.0 software (La Jolla, CA, USA). 

All data are shown as mean ± SEM. Trajectory data from four different groups (Figure 1B) were analyzed by repeated two-way ANOVA. The significant effects (main effects from two-way ANOVA) are presented as: ### *p*<0.001. To investigate the significant differences between groups in a single timepoint (in case of BW) or all other four groups analyzed by two-way ANOVA. The significant effects (main effects from two-way ANOVA) are presented as: ### *p*<0.001. The significant difference between groups analyzed by the post-hoc analysis, are presented as: *** *p*<0.001. In the case of two groups’ analysis, we have used a Student *t*-test (Figure 5A,B)

## 5. Conclusions

Our results indicate evidence correlating mitochondrial dysfunction with ASD and improved mitochondrial function following treatment with the KD. We demonstrated that the KD improves mitochondrial function and uncovered potential pathways through which the KD may alter brain mitochondrial morphology in BTBR mice. However, further investigation is required to understand the exact mechanism by which the KD may be improving mitochondrial morphology and function in BTBR mice.

While the KD is a readily available therapy, it is not without potentially significant side-effects [80]. In this regard, the fact that β-hydroxybutyrate alone was sufficient to promote hyperfused mitochondrial networks in primary neuronal cultures is exciting, as it suggests that the treatment with β-hydroxybutyrate alone (e.g., with ketone esters [81]), rather than the full KD, may be sufficient to improve mitochondrial function and ASD symptoms. Thus, understanding how the KD affects mitochondrial bioenergetics and dynamics is important in ultimately developing novel and improved therapeutic strategies for patients with ASD, especially those that mitigate the disabling core symptoms of the disorder.

## Figures and Tables

**Figure 1 ijms-21-03266-f001:**
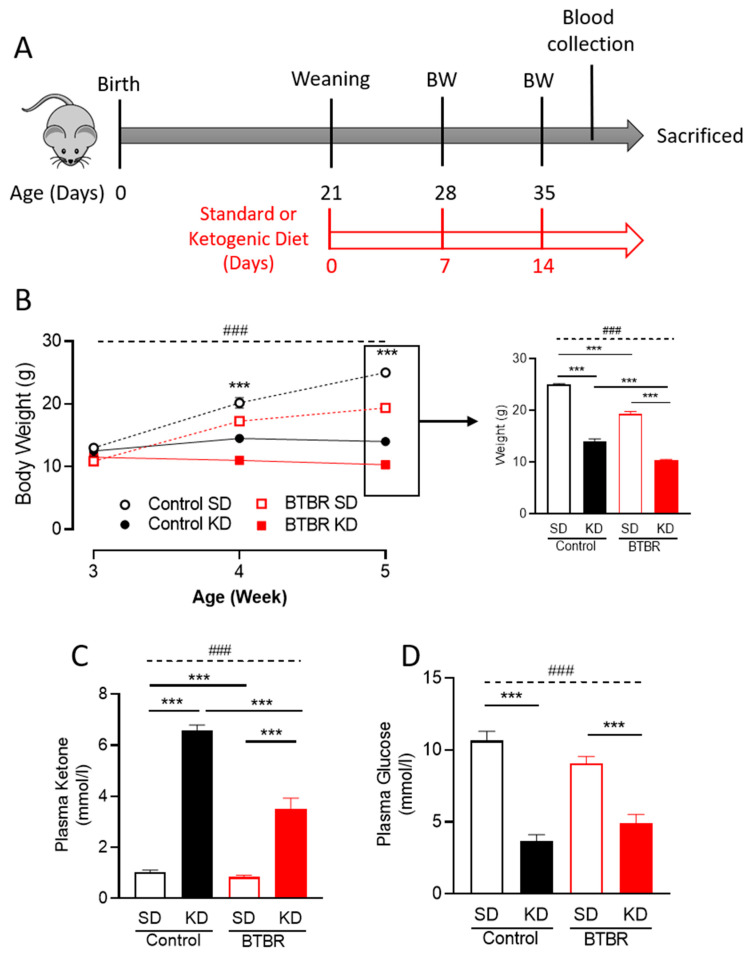
The ketogenic diet (KD) reduces body weight and induces ketosis in both control and BTBR mice. (**A**) Schematic drawing of the experimental protocol; after birth of control or BTBR mice, they were kept with their parents with a standard diet. After weaning at postnatal day 21 (PD21), the mice were placed on either a standard or a ketogenic diet. Body weight was measured after 7 and 14 days of diet (PD28 and PD35 weeks of age). Blood was collected to analyze for glucose and circulating ketone bodies. All mice were sacrificed at PD35 (after 2 weeks of diet intervention). (**B**) Average body weight trajectory of each group in response to the indicated diet (left panel). Data are shown as mean ± SEM, *n* = 4–8 per group. Data were analyzed by repeated two-way ANOVA. The significant effects (main effects from two-way ANOVA) are presented as: ### *p* < 0.001. Further, the significant differences between groups in each timepoint revealed by the post-hoc analysis are presented as *** *p* < 0.001. To explore the body weight changes at PD35 (after two-week diet intervention), single timepoint body weight difference data is presented (right panel). (**C**) Blood ketone and (**D**) glucose levels were measured in both control and BTBR mice sacrificed following the two-week diet intervention. Data are shown as mean ± SEM, *n* = 4–6 per group. Data were analyzed by two-way ANOVA. The significant effects (main effects from two-way ANOVA) are presented as ### *p* < 0.001. Further, the significant differences between groups revealed by the post-hoc analysis are presented as *** *p* < 0.001. BW: body weight and SD: standard diet.

**Figure 2 ijms-21-03266-f002:**
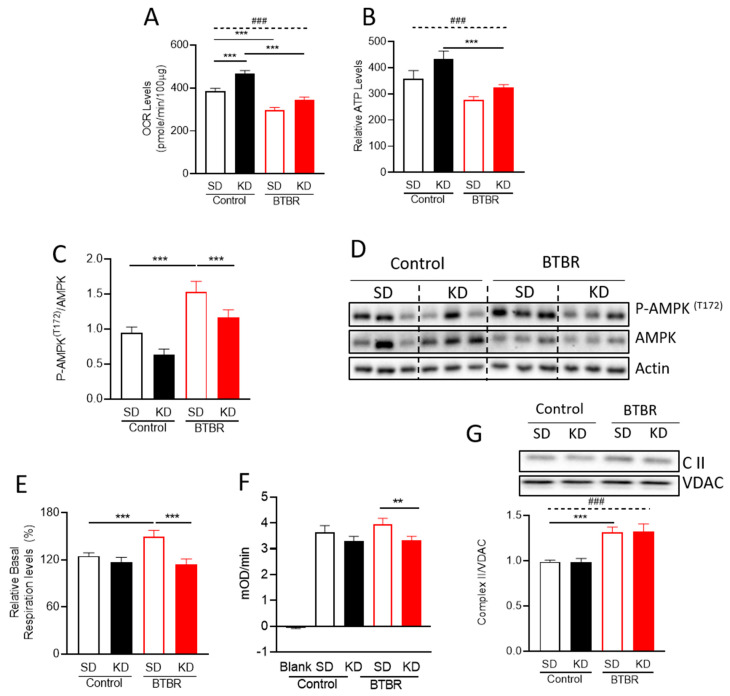
The ketogenic diet (KD) reverses alterations in mitochondrial bioenergetics in BTBR mice. Isolated mitochondria were used for mitochondrial bioenergetics assays with a Seahorse XF24 analyzer. (**A**) The bar graphs depict the oxygen consumption rates (OCR) under basal conditions and (**B**) relative ATP production using pyruvate and malate as substrates. (**C**) Densitometry analysis of P-AMPK ^T172^. (**D**) Representative blots for the densitometry analysis in Figure 2C. (**E**) Relative basal respiration levels via Complex II, after the addition of ADP (after subtraction of antimycin A-insensitive OCR (non-mitochondrial respiration)). The enzyme activity (**F**) and protein expression levels (**G**) of electron transport chain Complex II. Data are shown as mean±SEM, n = 6 per group. Data were analyzed by two-way ANOVA. The significant effects (main effects from two-way ANOVA) are presented as: ### *p*<0.001. Further, the significant differences between groups revealed by the post-hoc analysis, are presented as: *** *p*<0.001; ** *p*<0.01.

**Figure 3 ijms-21-03266-f003:**
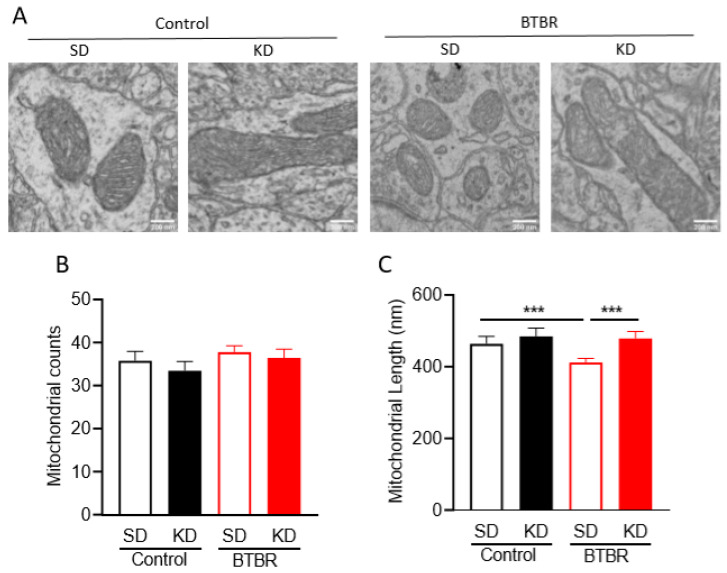
Transmission electron microscopy (TEM) data of neocortical tissue sections from BTBR and control mice. (**A**) Representative TEM images from control and BTBR mice fed SD or KD. Scale bars represents 200 nm. (**B**) Mitochondrial counts from micrographs of neocortical tissue sections from control and BTBR mice with/without the KD. (**C**) Mitochondrial length (nm) in TEM micrographs. Data are shown as mean ± SEM, *n* = 3 per group. Data were analyzed by two-way ANOVA. The significant differences between groups revealed by the post-hoc analysis are presented as *** *p* < 0.001.

**Figure 4 ijms-21-03266-f004:**
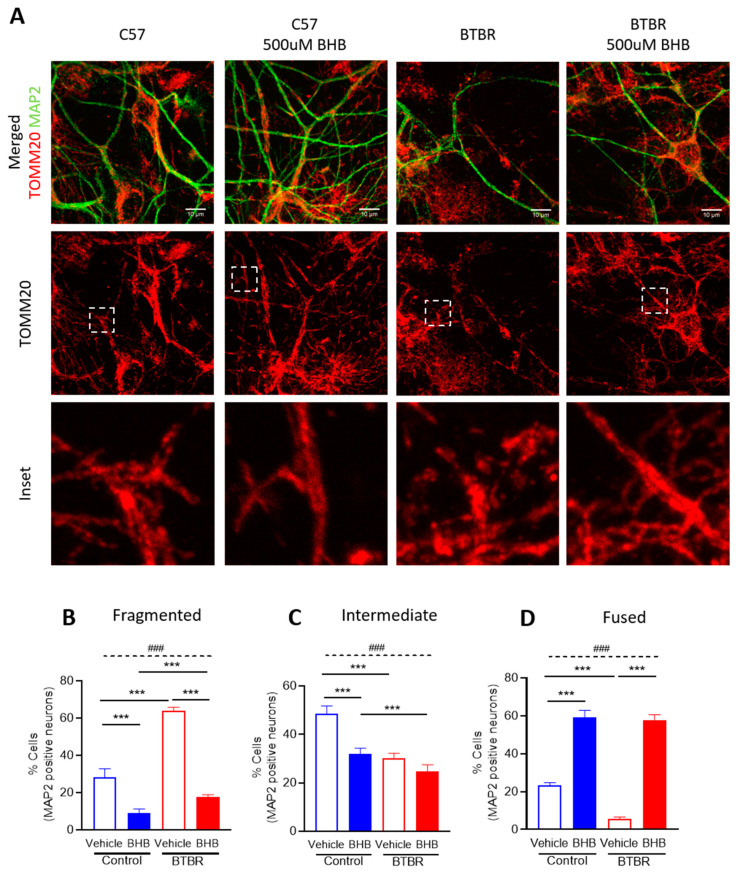
Mitochondrial morphology analysis in primary neuronal cultures from control and BTBR mice. (**A**) Representative confocal images of cortical neuronal cultures stained with antibodies against MAP2 (neurons, green) and TOMM20 (mitochondria, red). Dashed box highlights mitochondrial network in neurites (dendrites). Quantification of mitochondrial morphology (**B**) fragmented, (**C**) intermediate, and (**D**) fused in neurites in control and BTBR cultures from three independent replicates, treated as indicated. Data are shown as mean ± SEM. Data were analyzed by two-way ANOVA. The significant effects (main effects from two-way ANOVA) are presented as ### *p* < 0.001. Further, the significant differences between groups revealed by the post-hoc analysis are presented as *** *p* < 0.001.

**Figure 5 ijms-21-03266-f005:**
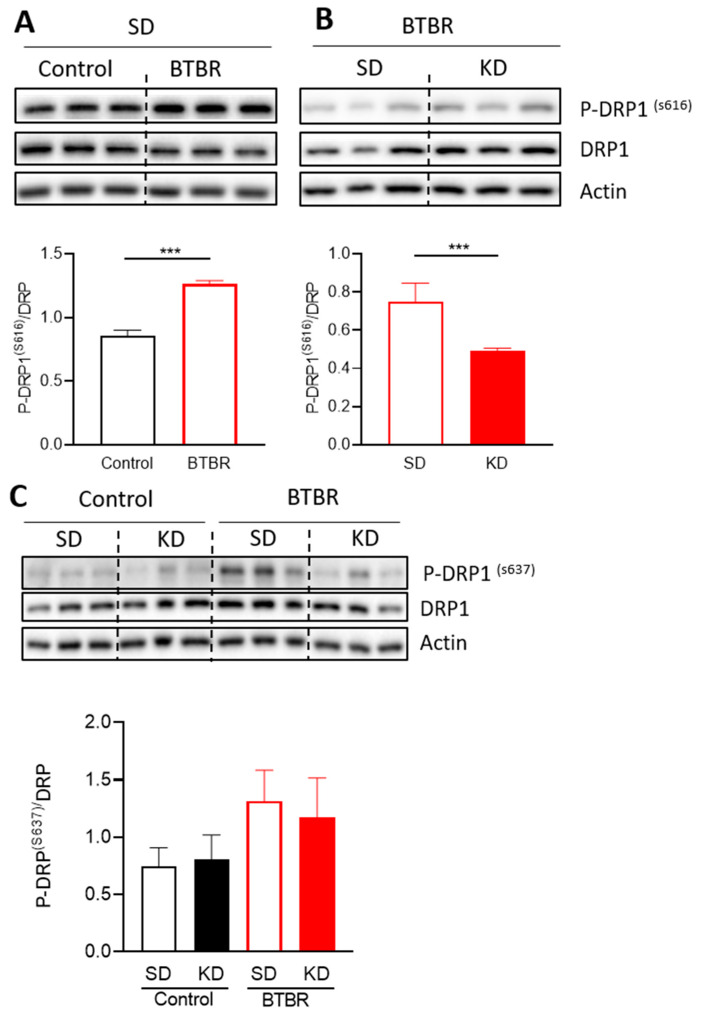
Changes in mitochondrial proteins responsible for mitochondrial fission and fusion due to the KD. Densitometry analysis and representative blots of P-DRP1 ^S616^, (**A**) comparing control and BTBR mouse under SD and (**B**) comparing BTBR mouse with or without KD. Data were analyzed by using a *t*-test; the significant differences between the groups are presented as *** *p* < 0.001. (**C**) Densitometry analysis and representative blots of P-DRP1 ^S637^. Data are shown as mean ± SEM, *n* = 6 per group. Data were analyzed by two-way ANOVA. The significant effects (main effects from two-way ANOVA) are presented as ### *p* < 0.001. Further, the significant differences between groups revealed by the post-hoc analysis are presented as *** *p* < 0.001.

**Figure 6 ijms-21-03266-f006:**
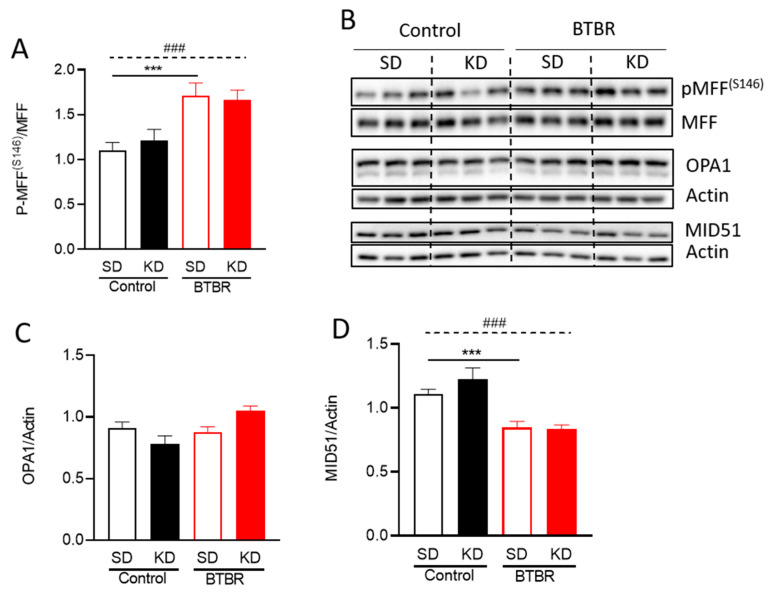
Changes in mitochondrial dynamic proteins due to KD. Densitometry analysis of (**A**) P-MFF ^S146^, (**C**) OPA1, and (**D**) MID51. (**B**) Representative blots for the densitometry analysis in (A,C,D). Data are shown as mean ± SEM, *n* = 6 per group. Data were analyzed by two-way ANOVA. The significant effects (main effects from two-way ANOVA) are presented as ### *p* < 0.001. Further, the significant differences between groups revealed by the post-hoc analysis are presented as *** *p* < 0.001.

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
