# Peer review of "Aberrant Mitochondrial Morphology and Function in the BTBR Mouse Model of Autism Is Improved by Two Weeks of Ketogenic Diet"

_ijms, 2020, doi:10.3390/ijms21093266_

Round 1

Reviewer 1 Report

Ahn et al investigate the effect of the ketogenic diet on the mitochondrial function in a mouse model of autism spectrum disorder (ASD), BTBR inbred mice. They found that early-life ketogenic diet in BTBR mice could improve mitochondrial function demonstrated by multiple experiments. The linkage between ASD and mitochondrial dysfunction is still not clear. The studies of the mitochondrial dysfunction in BTBR mice are still not many. Therefore, this study provides a novel direction of scientists in the ASD field to consider this direction. Overall, this study is interesting and novel. Below are my concerns regarding this study.

Line 42-43: The citation for the prevalence needs to be updated.

Line 85-87: The bodyweight dropped more than 20% which is not a reasonable range for diet change. It also not a common range for IACUC. The health issue is concerned about these KD-treated mice. How was the mortality rate, basic health condition, food intake, water intake, litter size, mouse locomotion in those KD-treated mice? Sometimes the mice need to adapt to the diet for a period of time to reach a stabilized condition.

Line 89: The ketone appears to be lower in BTBR compared to B6 in KD treatment groups, but not seemed in the SD treatment group.

Line 90: Has anyone reported the low glucose level after KD treatment at 5 weeks of age? Is the glucose level (<5 gm/dl)  high enough to maintain their health?

N number is low. For some of the groups, there were only 4 mice per group.

The rationale to test the age of 5 weeks for the testing of mitochondria function is unclear. The author will need to justify the experimental design in the text. For example, is the mitochondrial function.

Fig. 2. It appears that KD did not change mitochondria metabolism based on panel A-B. 

For Fig. 2, perhaps the altered mitochondria function by KD in BTBR mice is diluted by total brain measurement. Have authors ever look at the region-specific mitochondria metabolism.

Supplementary fig 1 should be moved to the main figure 2 to complement the result.

The description of Figure 4 can be elaborated more in section 2.5. Otherwise, it is hard to understand the interpretation of Fig. 4. For example, give a clear indicator or description on the fragmented, intermediate, fused in Fig. 4A inset.

Line 176: Provide the reference of BHB is the metabolite produced by KD.

No ASD behavior testing was done in this study which is a pity not showing the improvement in ASD behavior by KD.

Author Response

We thank the reviewers for their constructive criticism related to our paper. We hope that we have addressed all of the issues raised by the reviewers in the revised manuscript adequately. Related references for this response to the reviewers are listed at the end of this document following the MDPI-IJMS reference style.

Response to Reviewer #1:

  1. Ahn et al investigate the effect of the ketogenic diet on the mitochondrial function in a mouse model of autism spectrum disorder (ASD), BTBR inbred mice. They found that early-life ketogenic diet in BTBR mice could improve mitochondrial function demonstrated by multiple experiments. The linkage between ASD and mitochondrial dysfunction is still not clear. The studies of the mitochondrial dysfunction in BTBR mice are still not many. Therefore, this study provides a novel direction of scientists in the ASD field to consider this direction. Overall, this study is interesting and novel. Below are my concerns regarding this study.

Response: We thank the reviewer for their appreciation of this work and its contribution to this important emerging area of research. 

  1. Line 42-43: The citation for the prevalence needs to be updated.

Response: As requested, we have modified the text and now added a new updated reference.

Insert, page 2, paragraph 1, line 43-44

“A recent study suggested a 2.5% prevalence of ASD in the United States [1]”

  1. Line 85-87: The bodyweight dropped more than 20% which is not a reasonable range for diet change. It also not a common range for IACUC. The health issue is concerned about these KD-treated mice. How was the mortality rate, basic health condition, food intake, water intake, litter size, mouse locomotion in those KD-treated mice? Sometimes the mice need to adapt to the diet for a period of time to reach a stabilized condition.

Response: It is also unclear to us what the reviewer would consider a ‘reasonable range’. We are simply reporting the weight of the animals. Notably, weight loss is a well-established consequence of the KD. All animal procedures were carried out in accordance with the Canadian Animal Care Committee and approved by the University of Calgary Animal Care Committee. Importantly, the animals were closely monitored by veterinary technicians and there was no mortality or health concerns reported throughout the duration of the experiment.

As there were no issues in animal welfare in previous work, we did not measure food or water intake, nor animal locomotion. Regarding litter size, as we have mentioned in our method section that a BTBR mice breeding colony has been maintained at the Health Sciences Animal Resource Center at University of Calgary, and animals were picked randomly at weaning for experiments.

We agree with the reviewer that mice may take time to adapt to a diet, however, in this particular experiment, the 2 weeks KD was based primarily on the basis that we wanted to look at mitochondrial function in the same timeframe where we and others had previously seen behavioral benefits [2-4]. This reasoning has been clarified at the start of the results section.

Insert, page 2, paragraph 1, line 43-44

“As short term (2-3 weeks) administration of the KD improves ASD behaviors in BTBR mice [2,5], we decided to replicate previous KD treatments, but instead focus here on mitochondrial function.”

  1. Line 89: The ketone appears to be lower in BTBR compared to B6 in KD treatment groups, but not seemed in the SD treatment group.

Response: Yes, we agree. There is no reason that the basal ketone levels would be different between the BTBR and B6 mice on a standard diet.

  1. Line 90: Has anyone reported the low glucose level after KD treatment at 5 weeks of age? Is the glucose level (<5 gm/dl)high enough to maintain their health?

Response: We did not find any existing reports looking at glucose levels under these conditions. Lowering glucose level as well as weight loss with KD is a well-established concept [6-10]. The ‘low’ glucose levels are not a concern for the health of the animals, as they will have switched their metabolism to use ketones rather than glucose. It is this metabolic switch that is thought to alter mitochondrial function.

  1. N number is low. For some of the groups, there were only 4 mice per group

Response: We thank the reviewer for their concern however, we have enough sample numbers (n=8). Due to experimental error, sample loss in few cases we had to use 4 data points. Most cases we had 6-8 data points. Also, all the data were normally distributed and were with enough power to analyse the data.  

  1. The rationale to test the age of 5 weeks for the testing of mitochondria function is unclear. The author will need to justify the experimental design in the text. For example, is the mitochondrial function.

Response: We apologize for the confusion about the experimental design. There are two reasons for testing the mice at 5 weeks of age. First, it has been previously published that short-term administration of KD at this age had beneficial effect on the ASD behaviors of the BTBR mouse. Secondly, given ASD is a developmental disorder that affects children, we wanted to examine the effects on BTBR mouse at an early age. As discussed above, we included a more detailed rationale for the experimental design at the beginning of the results section.  

  1. 2. It appears that KD did not change mitochondria metabolism based on panel A-B.

For Fig. 2, perhaps the altered mitochondria function by KD in BTBR mice is diluted by total brain measurement. Have authors ever look at the region-specific mitochondria metabolism.

Response: We respectfully disagree with the notion that the KD does not change mitochondrial metabolism. There is a significant increase of basal oxygen consumption in control B6 mice. Moreover, even though the KD only shows a trend towards improvement in the BTBR mice, the respiration in BTBR on the KD is not distinguishable from the control B6 on the SD, suggesting improved function. In addition, the subsequent assays show further evidence for the KD changing mitochondrial metabolism.

One possible explanation for why we don’t see a significant change in BTBR mice on the KD compared to the SD, could certainly be due to regional differences, as suggested by the reviewer. Unfortunately, we did not investigate region-specific mitochondrial metabolism, as this would require even more mice and was beyond the scope of the current project. However, the notion of regional differences certainly intriguing and worthy of further study.

  1. Supplementary fig 1 should be moved to the main figure 2 to complement the result. (working on it)

Response: We agree with the reviewer and we have now combined the old supplementary figure 1A, 1B and 1C to figure 2.

  1. The description of Figure 4 can be elaborated more in section 2.5. Otherwise, it is hard to understand the interpretation of Fig. 4. For example, give a clear indicator or description on the fragmented, intermediate, fused in Fig. 4A inset.

Response: We thank the reviewer for their suggestion. We have now added a new Supplementary figure 2 with representative images (see updates sup figures in revised manuscript), and altered the text in order to clarify how mitochondrial morphology was analysed.

The updated result section is now reads –

Insert, page 7, paragraph 1, line 199-200

“Mitochondrial morphology in neurites (dendrites) was classified into fragmented, intermediate and fused networks as described in supplementary figure 2.”

Insert, supplementary figure 2 legend

“Supplementary Figure 2: Representative confocal images of mitochondrial networks in axonal neurites used as a reference to classify mitochondrial networks in primary neuronal cortical cultures from control and BTBR mice. Quantification of mitochondrial morphology was performed manually by examining at least 50 cells per condition and categorizing them into one of the following classes of mitochondrial network morphology: fragmented (small round puncta), intermediate (a mixture of tubulated networks and small puncta) or fused (primarily elongated tubules). The analyses were performed on 3 independent replicates for each condition.”

  1. Line 176: Provide the reference of BHB is the metabolite produced by KD.

Response: As requested, we have added references.

  1. No ASD behavior testing was done in this study which is a pity not showing the improvement in ASD behavior by KD.

Response: Our primary research question in the submitted manuscript was to look at mitochondrial function, as previous reports from us and others have already published the link between the KD and improved behaviours.

Insert, page 2, paragraph 5, line 83-85

In addition, preclinical studies have shown that the KD reduces ASD behaviors in multiple rodent models of ASD [5,11-14], including BTBR mice [4].

Reviewer 2 Report

The work by Ahn et al evaluates the effect of a ketonic diet in the BTBR mouse model of ASD. The compare the mitochondrial morphology and function of brain mitochondria obtained from either control (BL6) or autistic model (BTBR mice), that have been fed with either a normal diet (SD) or a ketonic diet (KD) during 2 weeks (from P21 till P35). 

Comments

1 The mouse model BTBR-KD do not show any increase in body weight during the 2 weeks of KD treatment (Figure 1b), that it is different to the BL6-KD model. Taking into account that this is a quite important growth phase in mice (from P21 till P35), it would be convenient to show that the animals are healthy. Moreover, due to this lack of growth, it is possible that there is an increase in the cell recycling activity (autophagy in general and mitophagy specifically) at a cellular level, that could also reduce the ROS production and, hence, improve the general performance of these animals. Did the authors check for this possibility? At least, this should be included in the discussion.

2 Although the authors make an effort to link their results with ASD, in this paper they do not show that indeed, the feeding protocol improves any of the aberrant behaviors of the autistic related phenotypes of the BTBR mice.

3 In general, the authors show the results as mean+/-SEM. However, the n numbers that have been used are relatively small (3-6 different samples). When so small populations are used, SD should be used, instead of SEM.

4 The text should be more precise when concerning to the samples that have been used for analysis, and do not lead to misunderstandings. For instance, in section 2.3, the authors talk about the mice, when they are indeed analyzing the mitochondrias than they have isolated from the indicated mice.

5 In figure 3, the authors show representative images of cerebral cortical mitochondria of the different mice models. Did they also observe a variation in the morphology of the crista of the mitochondria? Are they more tubular and less laminar? In the provided images, this is not easy to distinguish. Moreover, I recommend including more representative images for the quantifications included in figures 3B and 3C. I would also recommend to include the size on the scale bars in the figure legend (it is hardly visible as it is right now), and in figure 3B mitochondrial counts should be changed by mitochondrial density (including the size of the analyzed area).

6 In the Figure 4, in the provided images, it is hard to distinguish between the fragmented, intermediate and fused phenotypes that are quantified. I would recommend to show representative images of these three phenotypes. Moreover, as Figures 4B,C,D are related, the same scale in the y-axis should be used in all of them.

7 The explanations and results regarding Mff phosphorylation are really confused. The talk about two phosphorylation sites (S155 and S172), they show in figure 6 they show another residue (S146), in the discussion they say that they have shown three phosphorylation sites… Please, correct all these inconsistencies.

8 Their explanation linking the OPA1 levels with the morphology of the mitochondria has no so much sense, as there is not a correlation for all 4 conditions.

9 The discussion should be down tuned a bit. Sentences like “this study elucidates a mechanistic and functional link…” I would recommend to change it for “this study proposes a mechanistic and functional link…”.

Minor comments

Figure 1D check the unities of the y-axis

Figure 4: in the figure legend, it is wrongly indicated that MAP2 labels the axonal projections of the neurons. MAP2 is indeed a dendritic marker. This should be corrected.

Sentence 213: …these to… should be …these two…

Figure 5A and 5B show different ratios in phospho-DRP1 for BTBR-SD.

Please, check the quantification of Figure 5C. Based on the provided WB it is difficult to believe that phosphor-DRT1 in BTBR-KD is above 1.

Author Response

We thank the reviewers for their constructive criticism related to our paper. We hope that we have addressed all of the issues raised by the reviewers in the revised manuscript adequately. Related references for this response to the reviewers are listed at the end of this document following the MDPI-IJMS reference style.

Response to Reviewer #2:

The work by Ahn et al evaluates the effect of a ketonic diet in the BTBR mouse model of ASD. The compare the mitochondrial morphology and function of brain mitochondria obtained from either control (BL6) or autistic model (BTBR mice), that have been fed with either a normal diet (SD) or a ketonic diet (KD) during 2 weeks (from P21 till P35). 

 Comments

  1. The mouse model BTBR-KD do not show any increase in body weight during the 2 weeks of KD treatment (Figure 1b), that it is different to the BL6-KD model. Taking into account that this is a quite important growth phase in mice (from P21 till P35), it would be convenient to show that the animals are healthy. Moreover, due to this lack of growth, it is possible that there is an increase in the cell recycling activity (autophagy in general and mitophagy specifically) at a cellular level, that could also reduce the ROS production and, hence, improve the general performance of these animals. Did the authors check for this possibility? At least, this should be included in the discussion.

Response: We agree that this early stage growth is important. However, as we have mentioned in response to reviewer 1, there were no adverse effects on the overall health of the mice on the KD.

We did not measure any autophagy or mitophagy response due to KD. It is certainly possible that the KD could also affect these parameters. We have modified the discussion to include other mechanisms through, such as mitophagy, through which the KD may also have beneficial effects.

Insert, page 13, paragraph 1, line 361-365

“While we have seen that the KD improves mitochondrial morphology in BTBR mice, there are certainly other ways by which the KD could have beneficial effects on mitochondrial function. For example, the KD is known the reduce ROS by upregulating the NRF2 antioxidant pathway [15]. In addition, the KD could upregulate mitochondrial autophagy, which is an important mitochondrial quality control pathway [16].”

  1. Although the authors make an effort to link their results with ASD, in this paper they do not show that indeed, the feeding protocol improves any of the aberrant behaviors of the autistic related phenotypes of the BTBR mice.

Response: We have already published this information previously as mentioned in the text below.

Insert, page 2, paragraph 5, line 76-77

In addition, preclinical studies have shown that the KD reduces ASD behaviors in multiple rodent models of ASD [5,11-14], including BTBR mice [4].

  1. In general, the authors show the results as mean+/-SEM. However, the n numbers that have been used are relatively small (3-6 different samples). When so small populations are used, SD should be used, instead of SEM.

Response: We thank the reviewer for their concern, however, n=3 was used only in the Figure 3, transmission electron microscopy; everywhere we have enough sample numbers (n=8). Due to experimental error, sample loss in few cases we had to use 4 data points, however for most cases we had 6-8 data points. Also, all the data were normally distributed and were with enough power to analyse.  

  1. The text should be more precise when concerning to the samples that have been used for analysis, and do not lead to misunderstandings. For instance, in section 2.3, the authors talk about the mice, when they are indeed analyzing the mitochondrias than they have isolated from the indicated mice.

Response: We have now indicated precisely what tissue/media from the specific group were changed in our result section in multiple places, those are now highlighted.

  1. In figure 3, the authors show representative images of cerebral cortical mitochondria of the different mice models. Did they also observe a variation in the morphology of the crista of the mitochondria? Are they more tubular and less laminar? In the provided images, this is not easy to distinguish. Moreover, I recommend including more representative images for the quantifications included in figures 3B and 3C. I would also recommend to include the size on the scale bars in the figure legend (it is hardly visible as it is right now), and in figure 3B mitochondrial counts should be changed by mitochondrial density (including the size of the analyzed area).

Response: We did not observe any overt changes in the cristae morphology of the mitochondria We did not do a detailed analysis of EM images as we are not aware of reliable methods for quantifying subtle changes in cristae morphology.

The mitochondrial density might be an interesting metric. However, such analysis is painstaking and time consuming, and we do not currently have the resources. Moreover, we do not feel that it would add anything substantial to the analysis, as we did not see any differences in the overall number of mitochondria.

As requested, we have added additional representative EM images (Supplemental Figure 3), and have added the size for the scale bar in the legend of figure 3.

  1. In the Figure 4, in the provided images, it is hard to distinguish between the fragmented, intermediate and fused phenotypes that are quantified. I would recommend to show representative images of these three phenotypes. Moreover, as Figures 4 B,C,D are related, the same scale in the y-axis should be used in all of them.

Response: We thank the reviewer for their suggestions. We have corrected the axes on the graphs, as we have added a new Supplementary figure 2 with the representative images.

  1. The explanations and results regarding Mff phosphorylation are really confused. The talk about two phosphorylation sites (S155 and S172), they show in figure 6 they show another residue (S146), in the discussion they say that they have shown three phosphorylation sites… Please, correct all these inconsistencies.

Response: We apologize for the confusion. Due to sequences differences between human and mouse MFF proteins, the numbering is not the same. S172 in humans corresponds to S146 in mice. The discussion has been modified to make it easier to understand.

  1. Their explanation linking the OPA1 levels with the morphology of the mitochondria has no so much sense, as there is not a correlation for all 4 conditions.

Response: The minor point that we were trying to make is that the trend of changes in OPA1 processing may contribute to the increased mitochondrial length in the BTBR mice on the KD (along with changes in DRP1). Although the KD does not seem to have the same effect on OPA1 levels in control mice, it is important to note that the mitochondria are different to begin with, which may explain why both strains don’t always respond the same way to the KD.

  1. The discussion should be down tuned a bit. Sentences like “this study elucidates a mechanistic and functional link…” I would recommend to change it for “this study proposes a mechanistic and functional link…”.

Response: As suggested, we have toned down the discussion to be less declarative.

Minor comments

  1. Figure 1D check the unities of the y-axis

Response: We apologise for the mistake. We have now updated the figures.

  1. Figure 4: in the figure legend, it is wrongly indicated that MAP2 labels the axonal projections of the neurons. MAP2 is indeed a dendritic marker. This should be corrected.

 Response: We thank the reviewer for catching this error. We have changed ‘axonal projections’ to ‘dendrites’ in the text.

  1. Sentence 213: …these to… should be …these two…

Response: Corrected

  1. Figure 5A and 5B show different ratios in phospho-DRP1 for BTBR-SD.

Response: These are two independent western blots, probed and imaged at different times under different conditions, so we cannot compare the relative ratios between the blots. However, we can compare the relative differences in the ratios within the same blot, as we have done.

  1. Please, check the quantification of Figure 5C. Based on the provided WB it is difficult to believe that phosphor-DRT1 in BTBR-KD is above 1.

Response: We have rechecked these values and they were correct. P-DRP1 in BTBR-KD is expressed slightly lower than BTBR-SD, so as total DRP, thus when we normalised with the total DRP it remains higher.

Uncategorized References

  1. Zablotsky, B.; Black, L.I.; Maenner, M.J.; Schieve, L.A.; Danielson, M.L.; Bitsko, R.H.; Blumberg, S.J.; Kogan, M.D.; Boyle, C.A. Prevalence and Trends of Developmental Disabilities among Children in the United States: 2009–2017. Pediatrics 2019, 144, e20190811.
  2. Newell, C.; Shutt, T.E.; Ahn, Y.; Hittel, D.; Khan, A.; Rho, J.M.; Shearer, J. Tissue specific impacts of a ketogenic diet on mitochondrial dynamics in the BTBRT+ tf/j mouse. Frontiers in physiology 2016, 7, 654.
  3. Newell, C.; Shutt, T.E.; Ahn, Y.; Hittel, D.S.; Khan, A.; Rho, J.M.; Shearer, J. Tissue Specific Impacts of a Ketogenic Diet on Mitochondrial Dynamics in the BTBR(T+tf/j) Mouse. Frontiers in physiology 2016, 7, 654, doi:10.3389/fphys.2016.00654.
  4. Ruskin, D.N.; Svedova, J.; Cote, J.L.; Sandau, U.; Rho, J.M.; Kawamura, M., Jr.; Boison, D.; Masino, S.A. Ketogenic diet improves core symptoms of autism in BTBR mice. PLoS One 2013, 8, e65021, doi:10.1371/journal.pone.0065021.
  5. Verpeut, J.L.; DiCicco-Bloom, E.; Bello, N.T. Ketogenic diet exposure during the juvenile period increases social behaviors and forebrain neural activation in adult Engrailed 2 null mice. Physiol Behav 2016, 161, 90-98, doi:10.1016/j.physbeh.2016.04.001.
  6. Okuda, T.; Morita, N. A very low carbohydrate ketogenic diet prevents the progression of hepatic steatosis caused by hyperglycemia in a juvenile obese mouse model. Nutrition & diabetes 2012, 2, e50-e50.
  7. Kennedy, A.R.; Pissios, P.; Otu, H.; Xue, B.; Asakura, K.; Furukawa, N.; Marino, F.E.; Liu, F.-F.; Kahn, B.B.; Libermann, T.A. A high-fat, ketogenic diet induces a unique metabolic state in mice. American Journal of Physiology-Endocrinology and Metabolism 2007, 292, E1724-E1739.
  8. Badman, M.K.; Kennedy, A.R.; Adams, A.C.; Pissios, P.; Maratos-Flier, E. A very low carbohydrate ketogenic diet improves glucose tolerance in ob/ob mice independently of weight loss. American Journal of Physiology-Endocrinology and Metabolism 2009, 297, E1197-E1204.
  9. Newman, J.C.; Covarrubias, A.J.; Zhao, M.; Yu, X.; Gut, P.; Ng, C.-P.; Huang, Y.; Haldar, S.; Verdin, E. Ketogenic diet reduces midlife mortality and improves memory in aging mice. Cell metabolism 2017, 26, 547-557. e548.
  10. Ma, D.; Wang, A.C.; Parikh, I.; Green, S.J.; Hoffman, J.D.; Chlipala, G.; Murphy, M.P.; Sokola, B.S.; Bauer, B.; Hartz, A.M. Ketogenic diet enhances neurovascular function with altered gut microbiome in young healthy mice. Scientific reports 2018, 8, 1-10.
  11. Ahn, Y.; Narous, M.; Tobias, R.; Rho, J.M.; Mychasiuk, R. The ketogenic diet modifies social and metabolic alterations identified in the prenatal valproic acid model of autism spectrum disorder. Dev Neurosci 2014, 36, 371-380, doi:10.1159/000362645.
  12. Castro, K.; Baronio, D.; Perry, I.S.; Riesgo, R.D.S.; Gottfried, C. The effect of ketogenic diet in an animal model of autism induced by prenatal exposure to valproic acid. Nutr Neurosci 2017, 20, 343-350, doi:10.1080/1028415X.2015.1133029.
  13. Ruskin, D.N.; Fortin, J.A.; Bisnauth, S.N.; Masino, S.A. Ketogenic diets improve behaviors associated with autism spectrum disorder in a sex-specific manner in the EL mouse. Physiol Behav 2017, 168, 138-145, doi:10.1016/j.physbeh.2016.10.023.
  14. Ruskin, D.N.; Murphy, M.I.; Slade, S.L.; Masino, S.A. Ketogenic diet improves behaviors in a maternal immune activation model of autism spectrum disorder. PLoS One 2017, 12, e0171643, doi:10.1371/journal.pone.0171643.
  15. Milder, J.B.; Liang, L.-P.; Patel, M. Acute oxidative stress and systemic Nrf2 activation by the ketogenic diet. Neurobiology of disease 2010, 40, 238-244.
  16. Wang, B.-H.; Hou, Q.; Lu, Y.-Q.; Jia, M.-M.; Qiu, T.; Wang, X.-H.; Zhang, Z.-X.; Jiang, Y. Ketogenic diet attenuates neuronal injury via autophagy and mitochondrial pathways in pentylenetetrazol-kindled seizures. Brain research 2018, 1678, 106-115.

Reviewer 3 Report

Mitochondria dysfunction has been implicated in psychiatric disorders, including schizophrenia, autism spectrum disorder (ASD), Attention-deficit/hyperactivity disorder (ADHD), etc. Mitochondria are the energy-producing organelles of the cell. Their functions determine the integrity of cellular activities. Ketone bodies converted from fatty acids can serve as alternative sources for energy production. However, how ketone affects mitochondrial function and morphology in ASD animal models are still unclear. This manuscript by Ahn et al. examined functional and morphological changes of neuronal mitochondria in the BTBR mouse model of ASD fed with the ketogenic diet (KD) and investigated the underlying molecular mechanisms. The KD consists of high fat and low carbohydrates. The KD reduced body weight and plasma glucose levels. Interestingly, the KD improved mitochondrial function in BTBR mice by decreasing AMPK activation and increasing oxygen consumption rates. The BTBR mice with KD showed enlarged mitochondrial size and numbers, indicating restored mitochondrial function. To further investigate the molecular mechanisms mediating mitochondrial beneficial effects of the KD. The authors treated primary BTBR neuronal cells with β-hydroxybutyrate (βHB), ketone bodies generated by fatty acid oxidation. Neurons treated with the βHB exhibited an elevated number of fused mitochondria. Decreased pDRP1-S616 and increased OPA1 fusion protein indicated potential molecular mechanisms of mitochondrial morphological changes. Overall, the manuscript is concise and clear. It can be further improved by addressing the following questions.

  1. In figure 2D, the authors showed decreased phosphorylated AMPK with the KD treatment in BTBR mice. In the control group, the western blot showed a variation of AMPK level. Did the KD affect the total AMPK expression level?
  2. The KD treatment reduced pDRP1-S616 with unchanged DPR1 recruiting proteins in BTBR mice. Which signaling pathway may involve in ketone related phosphorylation of DRP1? Is pDRP1-S616 critical for driving the mitochondrial fission or for binding with recruiting proteins?
  3. The authors explored molecular mechanisms of mitochondrial beneficial effects of the KD. Can you generate a schematic to summarize how ketone bodies impact mitochondrial fission and fusion proteins?

Author Response

Response to Reviewer #3:

Mitochondria dysfunction has been implicated in psychiatric disorders, including schizophrenia, autism spectrum disorder (ASD), Attention-deficit/hyperactivity disorder (ADHD), etc. Mitochondria are the energy-producing organelles of the cell. Their functions determine the integrity of cellular activities. Ketone bodies converted from fatty acids can serve as alternative sources for energy production. However, how ketone affects mitochondrial function and morphology in ASD animal models are still unclear. This manuscript by Ahn et al. examined functional and morphological changes of neuronal mitochondria in the BTBR mouse model of ASD fed with the ketogenic diet (KD) and investigated the underlying molecular mechanisms. The KD consists of high fat and low carbohydrates. The KD reduced body weight and plasma glucose levels. Interestingly, the KD improved mitochondrial function in BTBR mice by decreasing AMPK activation and increasing oxygen consumption rates. The BTBR mice with KD showed enlarged mitochondrial size and numbers, indicating restored mitochondrial function. To further investigate the molecular mechanisms mediating mitochondrial beneficial effects of the KD. The authors treated primary BTBR neuronal cells with β-hydroxybutyrate (βHB), ketone bodies generated by fatty acid oxidation. Neurons treated with the βHB exhibited an elevated number of fused mitochondria. Decreased pDRP1-S616 and increased OPA1 fusion protein indicated potential molecular mechanisms of mitochondrial morphological changes. Overall, the manuscript is concise and clear. It can be further improved by addressing the following questions.

Response: We thank the reviewer for their appreciation and constructive comments.

  1. In figure 2D, the authors showed decreased phosphorylated AMPK with the KD treatment in BTBR mice. In the control group, the western blot showed a variation of AMPK level. Did the KD affect the total AMPK expression level?

Response: We have now added updated figures. The analysis stays the same. A new panel in Figure 2 (Figure 2F) represent the P-AMPK/Total AMPK. However, the expression changes when normalized to the loading control for both P-AMPK and total AMPK are now analysed and added to the supplementary figure 1B and 1C. When we analysed total AMPK expression, we found an elevated AMPK expression only in the control group due to KD. We have also updated the result section.

Insert, page 4, paragraph 3, line 134-141

“Moreover, the KD significantly reduced phosphorylation of AMPKT172 (ratio) in BTBR mice, with a trend towards reduced AMPK phosphorylation in controls (Figure 2F). This result is consistent when pAMPKT172 expression was normalized to the actin loading control (Figure 2G, Supplementary Figure 1C). Thus, these results indicate that the KD restores aberrant activation of AMPK in BTBR mice, possibly by increasing ATP production. Interestingly, when we analyzed total AMPK expression, we found an elevated AMPK expression only in the control group due to KD (Figure 2G, supplementary figure 1B)”

  1. The KD treatment reduced pDRP1-S616 with unchanged DPR1 recruiting proteins in BTBR mice. Which signaling pathway may involve in ketone related phosphorylation of DRP1? Is pDRP1-S616 critical for driving the mitochondrial fission or for binding with recruiting proteins?

Response: We thank the reviewer for raising these questions. In this current study we investigated whether the KD affects mitochondrial bioenergetics and dynamics and examined known mitochondrial fusion and fission proteins. While we saw clear differences, at this stage we cannot make any conclusions as to the mechanisms through which the KD affects DRP1. We agree with the reviewers that it is important to understand the exact pathway that is affected by the KD. However, more in-depth experimentation is required to confirm the pathway. As such, future studies are certainly warranted.

We have now amended the conclusion section as follows.

Insert, page 15, paragraph 5, line 507-511

“We demonstrated that the KD improves mitochondrial function and uncovered potential pathways through which the KD may alter brain mitochondrial morphology in BTBR mice. However, further investigation is required to understand the exact mechanism by which the KD may be improving mitochondrial morphology and function in BTBR mice”

  1. The authors explored molecular mechanisms of mitochondrial beneficial effects of the KD. Can you generate a schematic to summarize how ketone bodies impact mitochondrial fission and fusion proteins?

Response: With respect to a model, given that we don’t know how the KD affects DRP1, we feel it is premature to include a figure at this stage.